# Body Mass Index Reduction and Selected Cardiometabolic Risk Factors in Obstructive Sleep Apnea: Meta-Analysis

**DOI:** 10.3390/jcm10071485

**Published:** 2021-04-02

**Authors:** Marta Stelmach-Mardas, Beata Brajer-Luftmann, Marta Kuśnierczak, Halina Batura-Gabryel, Tomasz Piorunek, Marcin Mardas

**Affiliations:** 1Department of Treatment of Obesity, Metabolic Disorders and Clinical Dietetics, Poznan University of Medical Sciences, Szamarzewskiego 84 Street, 61-569 Poznan, Poland; 2Department of Pulmonology, Allergology and Pulmonary Oncology, Poznan University of Medical Sciences, Szamarzewskiego 84 Street, 60-569 Poznan, Poland; bbrajer@ump.edu.pl (B.B.-L.); halinagabryel@wp.pl (H.B.-G.); t_piorun@op.pl (T.P.); 3Department of Oncology, Poznan University of Medical Sciences, Szamarzewskiego 84 Street, 61-569 Poznan, Poland; martakusnierczak@gmail.com (M.K.); marcin.mardas@ump.edu.pl (M.M.)

**Keywords:** biological markers, weight loss, apnea–hypopnea index, blood pressure

## Abstract

Although clinical studies have been carried out on the effects of weight reduction in sleep apnea patients, no direct link has been shown between weight reduction and changes in cardio-metabolic risk factors. We aimed to analyze changes in the apnea–hypopnea index and selected cardio-metabolic parameters (total cholesterol, triglycerides, glucose, insulin, blood pressure) in relation to the reduction in body mass index in obstructive sleep apnea patients. Medline, Web of Science and Cochrane databases were searched to combine results from individual studies in a single meta-analysis. We identified 333 relevant articles, from which 30 papers were assigned for full-text review, and finally 10 (seven randomized controlled trials and three nonrandomized studies) were included for data analysis. One unit of body mass index reduction was found to significantly influence changes in the apnea–hypopnea index (−2.83/h; 95% CI: −4.24, −1.41), total cholesterol (−0.12 mmol/L; 95% CI: −0.22, −0.01), triglycerides (−0.24 mmol/L; 95% CI: −0.46, −0.02), fasting insulin (−7.3 pmol/L; 95% CI: −11.5, −3.1), systolic (−1.86 mmHg; 95% CI: −3.57, −0.15) and diastolic blood pressure (−2.07 mmHg; 95% CI: −3.79, −0.35). Practical application of lifestyle modification resulting in the reduction of one unit of body mass index gives meaningful changes in selected cardio-metabolic risk factors in obstructive sleep apnea patients.

## 1. Introduction

Obstructive sleep apnea (OSA) is a recognized cardio-metabolic disorder affecting 53% of the middle-to-older age general population, and 36% of OSA subjects having exclusive positional sleep apnea can be treated with positional therapy [1,2]. OSA is characterized by repetitive partial or complete closure of the upper airway during sleep that results in hypoxemia and hypercapnia, is frequently associated with arousals, and leads to an increase in myocardial oxygen demand [3]. The sum of the number of apneas and the number of hypopneas per hour is described by the apnea–hypopnea index (AHI) [4]. The AHI defines four grades of OSA: mild (5.0–14.9), moderate (15.0–29.9) and severe (≥30.0 events per hour) [4].

Currently, continuous positive airway pressure (CPAP) therapy is the ‘gold standard’ treatment for OSA, with dietary interventions and physical activity promoting weight loss encouraged in obese OSA individuals [5]. An increase of body weight by 10% over time increases the AHI, on average, by 30%, whereas a 10–15% reduction in body weight can reduce the AHI by 50% [6]. It has been shown that the Mediterranean diet improves OSA regardless of CPAP use and weight loss, whereas body-mass reduction itself improves OSA severity and symptoms [7]. Recent results from the Sleep AHEAD Study confirmed that individuals with OSA and type 2 diabetes mellitus receiving intensive lifestyle intervention for weight loss had reduced OSA severity, related to changes in body weight, baseline AHI and intervention independent of weight change at 10 years [8]. Additionally, previously published data have shown that OSA can itself be associated with dyslipidemia, hypertension and impaired glucose (Glc) tolerance independent of obesity [9,10,11]. OSA increases the risk of heart failure by 140%, the risk of stroke by 60% and the risk of coronary heart disease by 30%, causes significant sleep disturbances, leading to excessive daytime sleepiness and fatigue, depression (21.8%), anxiety (16.7%), posttraumatic stress disorder (11.9%), psychosis (5.1%) and bipolar disorder (3.3%) [12]. Although differentiated lifestyle interventions were applied in OSA individuals [13,14,15,16,17], until now there is no consistent finding that directly translates the effectiveness of lifestyle modification, expressed as body mass index (BMI) reduction, to changes in cardio-metabolic risk factors.

Because of this fact, we aimed this study to analyze changes in the AHI and selected cardio-metabolic parameters (concentration of total cholesterol (TC), triglycerides (TG), Glc, insulin, systolic and diastolic blood pressure (SBP, DBP)) in relation to reductions in the BMI in OSA patients.

## 2. Materials and Methods

### 2.1. Search Strategy, Inclusion and Exclusion Criteria

The databases Medline, Cochrane Library and Web of Knowledge were searched for clinical studies carried out between 1958 and November 2020 that reported the effect of lifestyle modification on BMI and selected cardio-metabolic parameters as primary or secondary outcomes in individuals with OSA.

The search strategy was restricted to humans, English language and original articles. The search was based upon the following index terms and titles: #1, sleep or apnea or obstructive sleep apnea or obstructive sleep apnea or sleep-disordered breathing and #2, diet or dietary intervention or diet, fat-restricted or energy intake or energy reduction and #3, weight loss or weight and #4, insulin or insulin resistance or insulin-secreting cells or glucose or lipids or triglycerides or cholesterol, and not animals. The PRISMA Statement was followed [18].

Only studies run with patients suffering from OSA indicating the changes in BMI, AHI and selected blood parameters after lifestyle modification were included. Intervention studies (randomized controlled trial, RCT, and nonrandomized controlled study, NRS) were taken into consideration. The articles that did not meet inclusion criteria were excluded.

### 2.2. Data Extraction and Analysis

Relevant articles were identified by screening the abstracts, titles and full texts. The study selection process was performed by two independent researchers in parallel for each database, cross-checked by a third reviewer. For each full-text paper, information was extracted including general information (study title, authors, year, journal), study characteristics (study design, country, length of intervention), characteristics of studied population (number, nationality, demographic characteristics of participants), assessment methods (body weight measurement, Glc, insulin, TC, TG and AHI measurements) and type of outcome (BMI changes, changes in selected cardio-metabolic parameters). The relationships between BMI reduction and AHI, TC, TG, Glc and insulin changes respectively were described as mean difference per 1 unit of BMI reduction if the effect of lifestyle modification was linked with reductions or increases in the analyzed parameters.

To assess the study quality, the Cochrane risk of bias for RCTs was used. For NRS, a nine-point scoring system according to the Newcastle–Ottawa scale was applied [19], where a high-quality study was defined by a threshold of ≥7 points.

### 2.3. Statistical Approach

When possible, the recorded Glc, TC and TG concentrations were converted to mmol/L and insulin concentration to pmol/L in order to standardize the results. A meta-analysis was performed to combine the results of the individual studies. Data were analyzed using a random-effects model, which allowed for true effect variation between studies. The effect size of a study was investigated by calculating mean difference per unit of BMI reduction (treated as an objective measurement of body weight change) with a 95% confidence interval.

The heterogeneity of the sum of studies was tested for significance. As a measure for quantifying inconsistencies, I2 was selected [20]. The results of the meta-analysis were visualized using a forest plot, which illustrates the results of the individual studies and the summary effect. The analysis was performed with Review Manager (RevMan) V5.3 (the Nordic Cochrane Centre, the Cochrane Collaboration, Copenhagen, Denmark, 2014).

## 3. Results

### 3.1. Search Results, Studies and Population Characteristics

We initially identified 333 potentially relevant publications from which, after title search, 97 articles were included. After duplicate removal, 30 papers were assigned for full-text review and finally 10 articles were included for data extraction and analysis [4,12,13,14,15,16,17,21,22,23,24]. The process outline and workflow is presented in Figure 1.

The characteristics of clinical studies (randomized and nonrandomized) and populations are presented in Table 1 and Table 2, respectively. The population consists of 1069 individuals and was characterized by a mean baseline BMI of >29 kg/m^2^, a mean age of 35–70 [13] and a predominance of Caucasian ethnicity. The durations of interventions ranged from 4 week [13] to 24 week [24] (with observations up to 2 year) [16], based on lifestyle modification [4,12,13,14,15,16,17,21,22,23,24].

The dietary strategies leading to a decrease in energy intake were based on a reduced intake of fat or general caloric restriction [4,12,13,14,15,16,17,21,22,23,24]. After the intervention period, mean BMI values decreased up to 5 units [16,22] and AHI values up to 15 [16] when only diet was used (Table 1).

The quantitative meta-analysis revealed a significant decrease in AHI per 1 unit of BMI change (mean difference: −2.83/h; 95% CI: −4.24, −1.41; *p* < 0.00001, I2 = 95%) (Figure 2).

For each study, the square represents the point estimate of the effect. Horizontal lines join lower and upper limits of the 95% CI of this effect. The area of shaded squares reflects the relative weight of the study in the meta-analysis. Diamonds represent the subgroup mean difference and pooled mean differences. CI indicates the confidence interval (upper and lower limit) [4,12,13,14,15,16,17,21,22,23,24].

### 3.2. Changes in Selected Cardio-Metabolic Parameters during Lifestyle Modification in Relation to BMI Reduction

Details about the analyzed biomarkers, at baseline and at the end of the intervention, or mean differences between the concentrations, were reported in only six studies [12,13,14,15,16,17,24]. The changes in TC concentration were followed by only Barnes et al. [12] and Nerfeld et al. [16]. The changes in TG were analyzed in five studies [12,13,16,17,24], fasting Glc in four studies [12,13,16,17], fasting insulin in three studies [12,16,17] and blood pressure in all six included studies [12,13,14,15,16,17,24] (Table 2).

Mean decreases ranged from 0.26 mmol/L [24] to 0.67 mmol/L [13] for TG concentrations, from 0.2 mmol/L [16] to 0.83 mmol/L [13] for Glc levels, up to 66.7 pmol/L for insulin levels, with changes in BP being highly differentiated [12,13,14,15,16,17,24] (Table 2).

The meta-analysis revealed a significant association between changes in the following cardio-metabolic risk factors per 1 unit of BMI change: TC (mean difference: −0.12 mmol/L; 95% CI: −0.22, −0.01; *p* = 0.03, I2 = 0%), TG (mean difference: −0.24 mmol/L; 95% CI: −0.46, −0.02; *p* = 0.03, I2 = 92%) and fasting insulin (mean difference: −7.3 pmol/L; 95% CI: −11.5, −3.1; *p* = 0.0007, I2 = 0%) (Figure 3), as well as in SBP (mean difference: −1.86 mmHg; 95% CI: −3.57, −0.15; *p* = 0.03, I2 = 76%) and DBP (mean difference: −2.07 mmHg; 95% CI: −3.79, −0.35; *p* = 0.02, I2 = 90%) (Figure 4).

For each study, the square represents the point estimate of the effect. Horizontal lines join lower and upper limits of the 95% CI of this effect. The area of shaded squares reflects the relative weight of the study in the meta-analysis. Diamonds represent the subgroup mean difference and pooled mean differences. CI indicates the confidence interval (upper and lower limit) [12,13,16,17,24].

### 3.3. Subgroup Analyses

Different subgroup analyses were performed to evaluate the possible influences of BMI reduction on AHI and cardio-metabolic risk-factor changes (study duration, study design and type of lifestyle modification). Nevertheless, none of the subgroup analyses, with regards to AHI changes and cardio-metabolic risk factors, indicated significance (*p* > 0.05)—in some cases, due to the limited number of studies, analysis was not possible (Table 3).

### 3.4. Risk of Bias and Publication Bias

The risk of bias for RCTs [4,13,15,17,21,23,24] is summarized in Figure 5 indicating mainly low risk or unclear risk. A high risk of bias was recognized for allocation in one study only [21]. For NRS, the Newcastle–Ottawa scale was applied and the mean score was 7 [12,16,22].

A funnel plot did not suggest real evidence of a publication bias for changes in AHI per 1 unit of BMI change (Figure 6a). However, funnel plots for other cardio-metabolic risk factors did reveal asymmetry, despite only a few studies being outliers, suggesting evidence of publication bias (Figure 6b,c).

The summary diamonds at the bottom of the plot represent the summarized effects using fixed and random effects models, where the random effects estimates are considered the primary findings for this study, due to heterogeneity [4,12,13,14,15,16,17,21,22,23,24].

## 4. Discussion

Here we present the first review summarizing the results from clinical studies performed on OSA with a primary interest in changes of selected cardio-metabolic outcomes as results of BMI reduction after applying lifestyle modification. The findings of the conducted systematic review present the beneficial effects of lifestyle modification on changes in BMI in OSA patients with strong clinical implications for positive changes in cardio-metabolic risk factors (TC, TG, fasting insulin and BP).

It was imperative for the conducted review to show that OSA patients should be recognized as the core group with a prime interest in changing diet being the determinant of “metabolic health”. However, only limited data are available with regards to dietary behaviors in OSA. It has to be highlighted that lifestyle change is very unlikely in this group of patients; therefore, the likelihood of the positive effect in OSA patients could be even smaller. It has been shown by Fogelholm et al. [25] that patients with OSA are more likely to suffer from increased excessive daytime sleepiness, sedentary behaviors associated with more time to eat, an increase in appetite and a liking for high-fat food. Therefore, an estimated 60–70% of patients with OSA can be categorized as obese, with a BMI greater than 30 kg/m^2^ [26]. Increasing body weight may predict an increase in clinical indicators of OSA severity, such as the AHI, which measures respiratory events during sleep [26,27]. For data consistency, we have expressed the changes in analyzed parameters per unit of BMI reduction. As a result, the differentiated duration of included dietary interventions could not influence the interpretation of obtained results. As previously shown in a prospective cohort study conducted from 1989 to 2000, a 10% weight gain may predict an approximate increase of 32% in the AHI. In contrast, a 10% weight loss may predict a 26% decrease in the AHI [27]. As shown by our meta-analysis, an improvement in AHI of more than 2.8/h per unit of BMI reduction may have practical significance. According to data published by Tuomilehto et al. [28], sustained improvement in body weight reduction and AHI can also be observed in 2 year post-intervention follow-ups. In clinical practice, changes from severe to either moderate or mild OSA may improve quality of life and reduce sleepiness in OSA patients. Specifically, it is worth noting that obesity itself contributes to daytime somnolence independent of OSA [12]. In clinical practice, it is possible without great efforts to have patients consult with dieticians in a way that will result in successful weight loss with long-term body weight maintenance [29,30]. We did not assess the benefits from applied dietary intervention in subgroup analysis with regards to overweight or obese-status analyzed individuals, as individuals in both conditions were included in the study population at baseline. The application of a very-low-caloric diet in routine practice needs replication of intervention studies in large-scale cohort studies and involvement of experienced nutritionists. It seems that the energy density of food is a simple and effective measure to manage weight in obese individuals with the aim of weight reduction [31]. A diet with self-regulation of dietary intake seems to be given a prominent role in the strategy of successful long-term weight loss among the obese [30]. Nevertheless, this measure could be combined with behavior therapy and physical activity and tailored to the individual situation [30]. A previously published study [29], based on individualized dietary counseling for obese subjects, indicated the mean percentage of body weight changes can be as follows: in the 6th week—5.9%, in the 12th week—10.9% and in the 52nd week—9.7% (*p* < 0.0001). These data are similar to those published on the OSA group of patients. For example, de Melo et al. [32] has shown that even a one month application of a low-energy diet resulted in body mass reduction in OSA patients (−3.7 ± 2.0% for the low protein group: 0.8 g of protein/kg/day and −4.0 ± 1.5% for the high protein group: 1.6 g of protein/kg/day; *p* < 0.001). It was also confirmed that a long-term lifestyle modification program could be more effective in reducing BMI (−1.8 kg/m^2^, 6.0% of the initial BMI *p* < 0.001) in comparison to the usual care of OSA patients (−0.6 kg/m^2^, 2.0% of the initial BMI; *p* < 0.001) [33].

The prevalence of cardiovascular diseases is increased in patients with OSA, possibly related to dyslipidemia in these individuals [34]. The findings from a meta-analysis of Dong et al. [35] support the idea that moderate–severe OSA significantly increased cardiovascular risk, in particular, stroke risk. Although insulin resistance is also very often diagnosed in OSA patients, its contribution to the dyslipidemia of OSA remains unclear. Our analysis indicates statistically significant positive changes in TC, TG and fasting insulin per unit of BMI reduction, which was also confirmed in single studies after applied lifestyle modification [12,16,17,24]. However, the change in TC concentration was analyzed based solely on data from two studies, but with 0% of heterogeneity. Nevertheless, the observed particular change in TG concentration seems to be more valuable as a marker for the assessment of dyslipidemia in the current study. Taking into account the results obtained from a 16 week intervention, based on a low-caloric diet, in a group of obese women sufferers of dyslipidemia, we could expect changes in TG around 13% [36]. Therefore, obtaining a similar result when expressed per unit of BMI reduction is sufficient for OSA to reach a goal of complementary use of diet with medical treatment. Reductions in hepatic TG content are strongly associated with improved hepatic insulin sensitivity and lipoprotein metabolism through different mechanisms, including the effect of inflammatory intermediates on insulin receptor signaling and very-low-density lipoprotein synthesis [37]. In contrast, no effect on peripheral insulin resistance can be observed, which may support the hypothesis that a relatively small pool of intrahepatic lipids may be responsible for dysregulated hepatic Glc metabolism [38,39]. In our analysis, no significant change in Glc concentration was observed, which may suggest long-term dysregulation in Glc metabolism. Although, the overall effect of diet on SBP and DBP expressed per unit of BMI reduction was also significant (between 1.86 and 2.07 mmHg per unit of BMI reduction) in our study, it can be interpret as minor from a clinical point of view. Nevertheless, as reported by Tuomilehto et al. [17], a notable number of patients were able to discontinue drug treatment for hypertension, diabetes and hypercholesterolemia after lifestyle modification. It seems justified to consider the relationship between BMI reduction and changes in cardio-metabolic biomarkers in obese and overweight individuals when evaluating patients found to have OSA. It should be highlighted that, currently, more personalized attention to patients is present, and more focus on different phenotypes of OSA is recognized, such as REM-dependent phenotype or positional phenotype, which in future studies also should be considered when analyzing the effect of diet on cardio-metabolic factors [40,41]. Finally, a recent meta-analysis including 39 RCTs with 6954 subjects has shown that there is a risk of an increase in BMI in patients with OSA following CPAP treatment, especially in those with less than 5 h/night of CPAP use [42]. In this context, the greatest reduction in BP observed in the study by Chirinos et al. (Table 2) was when CPAP was combined with diet therapy [24], which further emphasizes the importance of lifestyle modifications, especially in obese individuals with OSA.

## 5. Limitations

Despite an increasing number of dietary intervention studies, the body of evidence remains limited by either small sample size (an inclusion-only arm of intervention without use of devices) or inclusion of nonrandomized trials and studies with behavioral support. Only future clinical trials with long-term follow-up periods can address this limitation. Moreover, no information on possible comorbidities of individuals in the analyzed studies was provided, which could also influence obtained data. Furthermore, the present findings are based on limited ethnicity (Caucasian); therefore, results could vary as a function of ethnic background. Although, the duration of the interventions in analyzed studies was relatively long (up to 24 weeks), we could observe long-term follow-up changes of analyzed cardio-metabolic parameters in a few studies. We did not analyze interventions based on physical activity, which is commonly recommended to OSA patients with standard therapy and applied as a pragmatic strategy (hard to accept by patients having stable behavior habits), though alone it does not provide significant clinical benefits [39]. It must be highlighted that only studies that could show the effectiveness of lifestyle modification (either diet alone or diet with physical activity) were included. Some studies published in the grey literature may have been missed by our literature search.

## 6. Conclusions

In conclusion, we find that lifestyle modification resulting in the reduction of one unit of BMI gives meaningful and positive changes in selected cardio-metabolic risk factors such as TC, TG, fasting insulin and BP in OSA patients. Broader interventional studies are needed to assess different dietary approaches in OSA individuals.

## Figures and Tables

**Figure 1 jcm-10-01485-f001:**
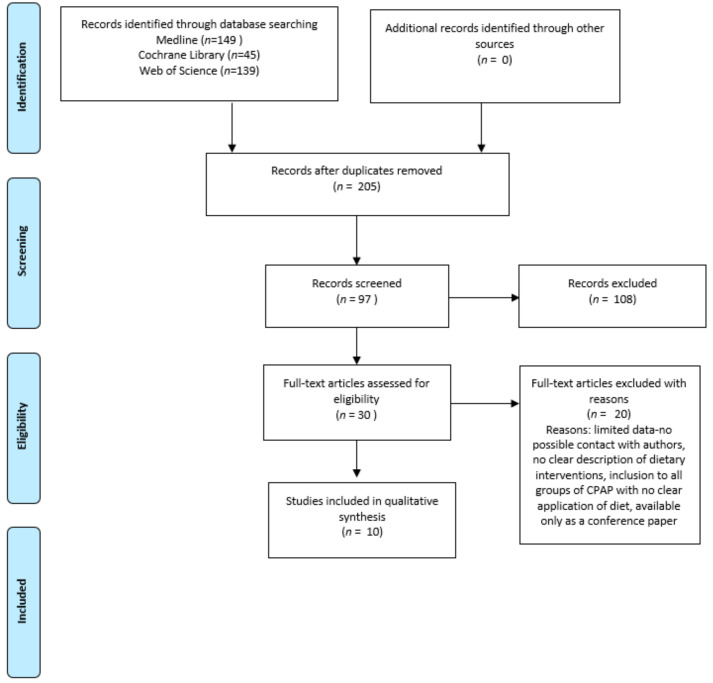
Process of the literature search for dietary intervention and cardio-metabolic risk factors in adults.

**Figure 2 jcm-10-01485-f002:**
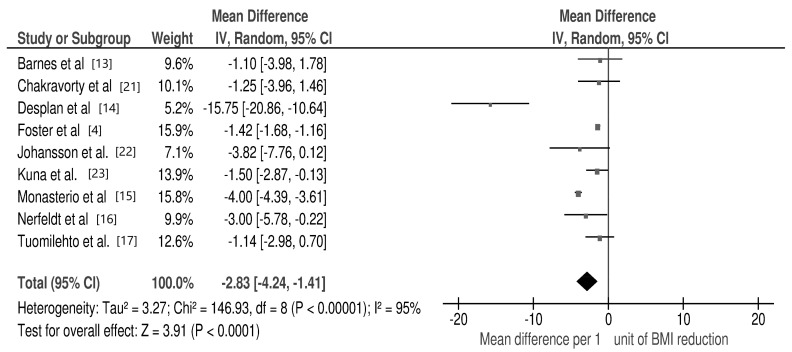
Forest plot for mean apnea–hypopnea index (AHI) change per 1 unit of body mass index (BMI) reduction in selected studies.

**Figure 3 jcm-10-01485-f003:**
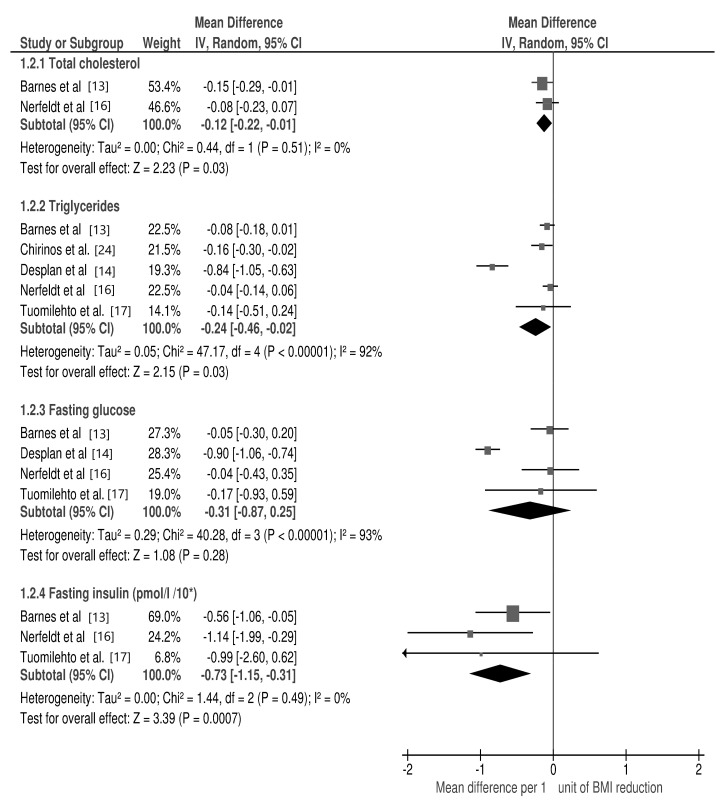
Forest plot for mean cardio-metabolic parameters change in subgroup analysis: total cholesterol, triglycerides, glucose, insulin per 1 unit of body mass index (BMI) reduction in selected studies.

**Figure 4 jcm-10-01485-f004:**
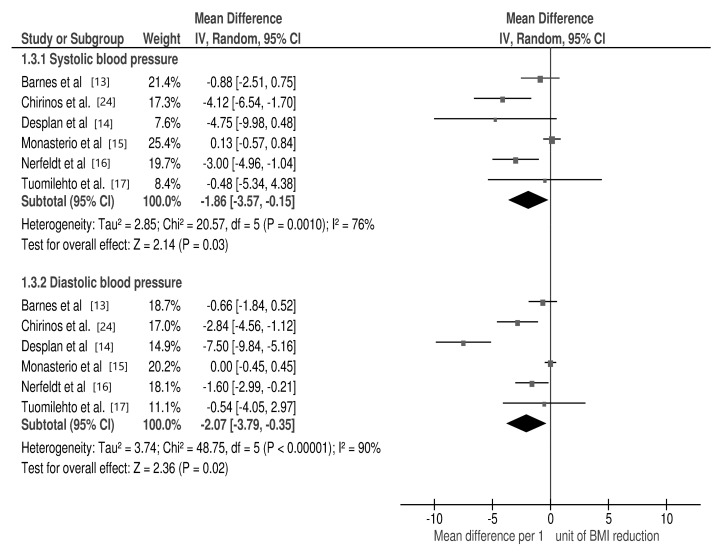
Forest plot for mean cardio-metabolic parameter changes in subgroup analysis: systolic and diastolic blood pressure per 1 unit of body mass index (BMI) reduction in selected studies.

**Figure 5 jcm-10-01485-f005:**
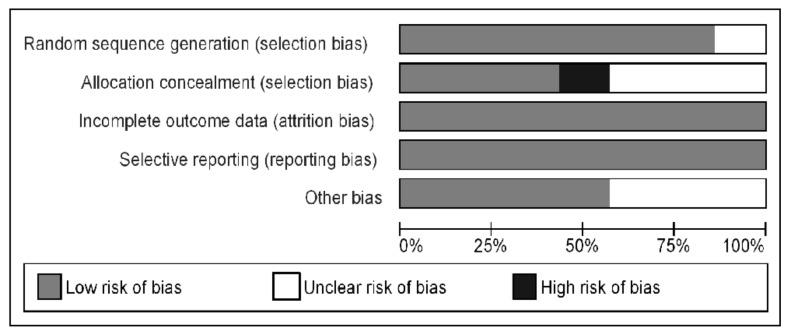
Risk of bias graph: review authors’ judgments about each risk of bias item presented as percentages across all included studies.

**Figure 6 jcm-10-01485-f006:**
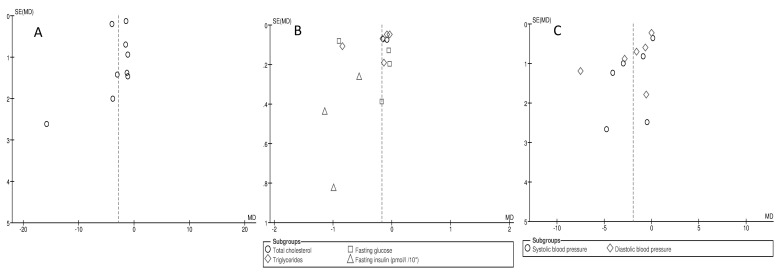
Funnel plot of standard error by standard differences in means of: (**A**) apnea–hypopnea index, (**B**) total cholesterol, triglycerides, glucose and insulin, (**C**) systolic and diastolic blood pressure in selected studies per unit of body mass index reduction.

**Table 1 jcm-10-01485-t001:** Characteristics of the included studies and changes in BMI and AHI during dietary intervention in the study and control groups.

Study	Study Design	Subjects (n) ^§^	Age (Years) Mean ± SD	% of Women	Nationality	Analyzed Groups	Intervention	Time of Intervention	BMI (kg/m^2^) Mean ± SD	AHI (A + H/h) Mean ± SD		
									Baseline	Intervention	Baseline	Intervention
Barnes et al. 2009 [13]	NRS	12	All: 42.3 ± 10.4	100	Australia	SG	VLED *	16 wk	36.1 ± 4.3	30.1 ± 4.2	24.6 ± 12.0	18.3 ± 11.9
Chakravorty et al. 2002 [21]	RCT	53	All: 49.0 ± 11.0	-	Great Britain	SG CG	CPAP Diet	12 wk	40.0 ± 14.5 32.5 ± 5.5	40.0 ± 12.5 31.7 ± 5.6	55.0 ± 28.7 35.0 ± 19.1	8.0 ± 28.0 34.0 ± 21.0
Chirinos et al. 2014 [24]	RCT	146	48.3 49.8 49.0	41.0 59.7 49.0	American	SG 1 SG 2 CG	Diet CPAP Diet + CPAP	24 wk	38.3 ±5.5 38.2 ±7.2 37.7 ±5.5	-	39.7 ± 20.3 41.2 ± 20.96 47.1 ± 26.86	-
Desplan et al. 2014 [14]	RCT	22	All: 35–70	-	France	SG CG	Diet + EAS * EAS	4 wk	29.9 ± 3.4 31.3 ± 2.5	29.1 ± 3.1 31.3 ± 2.2	40.6 ± 19.439.8 ± 19.2	28.0 ± 19.3 45.4 ± 22.5
Foster et al. 2009 [4]	RCT	264	All: 61.2 ± 6.5	-	American	SG CG	Diet * Education support	16 wk	−3.8 ± 0.3 −0.2 ± 0.3	−5.4 ± 1.5 4.2 ± 1.4		
Johansson et al. 2011 [22]	NRS	62	All: 48.7± 7.3	-	Sweden	SG	VLCD	9 wk	−5.5 ± 2.0		−21 ± 16	
Kuna et al. 2013 [23]	RCT	264	All: 61.3 ± 6.5	59	American	SG CG	Diet * Education support	16 wk	−3.8 ± 0.74 *** −0.2 ± 0.66 ***	−5.7 ± 1.5 4.0 ± 1.4		
Monasterio et al. 2001 [15]	RCT	142	54.0 ±9.0 53.0 ± 9.0		Spain	SG CG	Sleep hyg. + diet Sleep hyg. + diet + CPAP	12 wk	29.5 ±3.3 29.4 ±3.7	28.5 ±3.5 ** 29.5 ±2.8 **	21.0± 6.0 20.0 ± 6.0	17.0 ± 10.0 * 6.0 ± 8.0 *
Nerfeldt et al. 2010 [16]	NRS	33	All: 52(31–68)	27.2	Sweden	SG	VLCD + behavioral support	8-wk (observation: 24mo)	40.0 ± 5.0	35.0 ± 3.0	43.0 ± 24.0	28.0 ± 19.0
Tuomilehto et al. 2009 [17]	RCT	72	51.8 (9.0) 50.9 (8.6)	35	Finland	SG CG	VLCD + lifestyle modification lifestyle modification	12 wk (observation: 12mo)	−3.5 ± 2.1 −0.8 ± 2.0	−4.0 ± 5.6 0.3 ± 8.0		

* physical activity recommended, ** after 6-months,*** calculated, ^§^—the number that completed the study; SD—standard deviation; AHI—the apnea–hypopnea index; BMI—body mass index; CBT—cognitive-behavioral therapy; CG—control group; CPAP—continuous positive airway pressure; EAS—education activity session; NRS—nonrandomized controlled study; RCT—randomized clinical trial; SG—study group; VLCD—very-low-caloric diet, VLED-very low energy density.

**Table 2 jcm-10-01485-t002:** Mean changes in cardio-metabolic risk factors during dietary intervention in the study and control groups in selected studies.

Study	Analyzed Groups	TC (mmol/L) Mean ± SD		TG (mmol/L) Mean ± SD	Fasting Glc (mmol/L) Mean ± SD			Fasting Insulin (pmol/L) Mean ± SD		SBP/DBP (mmHg) Mean ± SD	
		Baseline	Intervention	Baseline	Intervention	Baseline	Intervention	Baseline	Intervention	Baseline	Intervention
Barnes et al. 2009 [13]	SG	5.3 ± 1.1	4.4 ± 1.0	1.5 ± 0.8	1.0 ± 0.6	6.1 ± 2.2	5.8 ± 1.5	107.6 ± 41.7	74.3 ± 32.6	125.8 ± 14.0/77.3 ± 10.6	120.5 ± 9.8/ 73.3 ± 6.3
Chirinos et al. 2014 [24]	SG 1 SG 2 CG	-	-	−0.26 (−0.49 to −0.03) ** −0.08 (−0.27 to 0.11) ** −0.60 (−0.86 to −0.34) **	-		-	-		−6.8 (−10.8 to −2.7)/ −4.7 (−7.7 to −1.7) ** −3 (−6.5 to 0.5)/ −3.5 (−6.1 to −0.9) ** −14.1 (−18.7 to −9.5)/ −10.6 (−14 to −7.2) **	
Desplan et al. 2014 [14]	SG CG	-	-	1.87 ± 1.01 1.70 ± 0.53	1.20 ± 0.29 1.72 ± 0.81	5.61 (4.94–6.17) * 5.44 (5.17–5.67) *	4.78 (4.5–5.44) * 5.17 (4.89–6.0) *	-	-	141.8 ± 15.8/ 79.0 ± 7.1 128.1 ± 14.4/ 82.5 ± 7.1	145.6 ± 23.2/ 73.0 ± 10.3 128.0 ± 18.2/ 78.8 ± 9.7
Monasterio et al. 2001 [15]	SG CG	-	-	-	-	-	-	-	-	132 ± 17/84 ± 11 126 ± 17 81 ± 12	131 ± 16/84 ± 10 126 ± 15/81 ± 10
Nerfeldt et al. 2010 [16]	SG	5.3 ± 1.1	4.9 ± 1.2	1.8 ± 0.8	1.6 ± 0.7	7.2 ± 2.7	7.0 ± 3.3	147 ± 78	90 ± 52	144 ± 19/89 ± 14	129 ± 10/81 ± 6
Tuomilehto et al. 2009 [17]	SG CG	-	-	−0.48 ± 1.13 −0.006 ± 0.65	−0.6 ± 2.3 −0.4 ± 1.4			−34.7 ± 48.6 8.33 ± 23.6		−1.7 ± 14.7/−1.9 ± 10.6 −1.1 ± 19.6/−0.4 ± 12.6	

* Median, 25%, 75%; ** 95% CI; CG—control group; DBP—diastolic blood pressure; Glc—glucose; SBP—systolic blood pressure; SG—study group; TC-total cholesterol; TG-triglycerides.

**Table 3 jcm-10-01485-t003:** Subgroup analyses for possible influences of BMI reduction on AHI and cardio-metabolic risk-factor changes (study duration, study design and type of dietary intervention).

Analyzed Parameter	Duration			Design			Intervention		
	Short-Term	Long-Term	*p*	RCT	NRS	*p*	LC Diet	VLCD	*p*-Value
AHI [A + H/h]	−7.24 (14.15, −0.34)	−1.87 (−3.37, −0.37)	0.14	−2.95 (−4.65, −1.26)	−2.44 (−4.22, −0.65)	0.68	−3.35 (−5.25, −1.46)	−1.81 (−3.09, −0.53)	0.19
TC (mmol/L)	-	-	-	-	-	-	-	-	-
TG (mmol/L)	−0.47 (−1.25, 0.31)	−0.11 (−0.19, −0.03)	0.37	−0.39 (−0.87, 0.10)	−0.08 (−0.18, 0.02)	0.22	−0.50 (−1.16, 0.17)	−0.06 (−0.13, 0.01)	0.21
fasting Glc (mmol/L)	−0.49 (−1.33, 0.35)	−0.06 (−0.30, 0.18)	0.34	−0.63 (−1.32, 0.06)	−0.05 (−0.26, 0.16)	0.11	-	-	-
fasting insulin (pmol/L)	-	-	-	-	-	-	-	-	-
SBP (mmHg)	−3.22 (−5.05, −1.38)	−1.22 (−3.05, 0.62)	0.12	−2.05 (−4.98, 0.89)	−1.87 (−3.94, 0.21)	0.92	−2.49 (−6.11, 1.14)	−1.70 (−3.29, −0.10)	0.70
DBP (mmHg)	−4.47 (−10.25, 1.31)	−0.68 (−1.41, 0.05)	0.20	−2.70 (−6.01, 0.60)	−0.87 (−1.63, −0.10)	0.29	−3.31 (−7.33, 0.71)	−1.02 (−1.89, −0.15)	0.28

AHI—the apnea–hypopnea index; DBP—diastolic blood pressure; Glc—glucose; LC—low caloric; SBP—systolic blood pressure; TC-total cholesterol; TG-triglycerides; NRS—nonrandomized controlled study; *p*—*p*-value; RCT—randomized clinical trial; VLCD—very-low-caloric diet.

## Data Availability

To get an access to secondary data please contact correspondence author.

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
