# Peer review of "Body Mass Index Reduction and Selected Cardiometabolic Risk Factors in Obstructive Sleep Apnea: Meta-Analysis"

_jcm, 2021, doi:10.3390/jcm10071485_

Round 1
Reviewer 1 Report
In this manuscript, the authors perform a meta-analysis of studies on weight loss in subjects with OSA to determine if there are changes in cardiometabolic risk factors. The meta-analysis shows a reduction in AHI with weight loss as well as improvements in cholesterol, triglyercides, fasting insulin and blood pressure measures.
Major Comments
- While the results are interesting, the authors have not attempted to link the changes to changes in AHI. In other words, are the improvements in the risk factors due to weight loss alone? Are they in proportion to change in AHI? This is important as we already know that weight loss improves these factors.
- Another limitation is that of CPAP. Were all the studies performed on patients NOT on CPAP?
- Discussion would be aided by comparison to studies of weight loss in general, not just weight loss in OSA. Are the changes comparable?
- Finally, why exclude studies on bariatric surgery?
Minor Comments
- In Table 1, the study by Kuna is listed as Australia but is an American study. Please make sure all of the countries are correct (I only checked that one as I know Dr. Kuna works at UPenn.
- Why does Table 2 only have 5 of the 10 studies? Or why only 5 of the 6 studies that are used to discuss changes in risk factors?
- Page 11 of manuscript, first paragraph near end: all of a sudden the authors bring up pulmonary HTN; not sure why as not otherwise discussed.
1.
Author Response
We would like to thank the reviewer for her/his thoughtful comments and efforts towards improving our manuscript. The point-to-point response was included below:
In this manuscript, the authors perform a meta-analysis of studies on weight loss in subjects with OSA to determine if there are changes in cardiometabolic risk factors. The meta-analysis shows a reduction in AHI with weight loss as well as improvements in cholesterol, triglyercides, fasting insulin and blood pressure measures.
Major Comments
While the results are interesting, the authors have not attempted to link the changes to changes in AHI. In other words, are the improvements in the risk factors due to weight loss alone? Are they in proportion to change in AHI? This is important as we already know that weight loss improves these factors.
Thank you for this comment. Forest plot for mean apnea-hypopnea index (AHI) change per 1 unit of body mass index (BMI) reduction in selected studies was presented – Fig.2 . Subgroup analyses for possible influences of BMI reduction on AHI and cardio-metabolic risk factors changes (study duration, study design and type of dietary intervention) was additionally presented in table 3.
Another limitation is that of CPAP. Were all the studies performed on patients NOT on CPAP?
Thank you for this comment. As was presented study characteristic (table 1) in 3 studies out of 10 patients were also randomized to one of the groups that received CPAP, however in meta-analysis this group were not considered.
Discussion would be aided by comparison to studies of weight loss in general, not just weight loss in OSA. Are the changes comparable?
Thank you for this comment. This information was added in the discussion section.
Finally, why exclude studies on bariatric surgery?
Thank you for this comment. That would be also interesting, however we were interested how diet modification influence OSA patients not surgery itself.
Minor Comments
In Table 1, the study by Kuna is listed as Australia but is an American study. Please make sure all of the countries are correct (I only checked that one as I know Dr. Kuna works at UPenn.
Thank you for this comment. I was corrected and all other studies were checked, whether there are no more mistakes.
Why does Table 2 only have 5 of the 10 studies? Or why only 5 of the 6 studies that are used to discuss changes in risk factors?
Thank you for this comment. As usually in case of SLR not all selected papers according to primary focus (BMI and AHI changes) include additional information on single cardio-metabolic parameters that are still additional point of interest.
Page 11 of manuscript, first paragraph near end: all of a sudden the authors bring up pulmonary HTN; not sure why as not otherwise discussed.
Thank you for this comment. It was deleted.
Reviewer 2 Report
The manuscript entitled „Body mass index reduction and cardiometabolic risk factors in obstructive sleep apnea reached by modified lifestyle: meta-analysis” provides information on available data regarding the influence of weight loss on AHI and chosen cardiometabolic parameters.
OSA is not a cardio-metabolic disorder – it is a sleep-breathing disorder affecting the cardiometabolic parameters of the individuals.
The prevalence of the disorder is out-of-date. Two references given for the prevalence do not provide only cite given prevalence. If original publication were considered by the authors, they would have seen that it has been published in 1993. Present data suggest that it affect a vastly greater population (see Heinzers’ and Sentras’ papers).
Please expanded on the weight loss effect on AHI in the introduction, or give additional references, as the given one (no. 6) is from 1985 and this study included 23 patients.
While discussing, the cardiometabolic risk factors in OSA, the frequency of its comorbidities from this group should be included.
Authors should undertake the topic that lifestyle change is very unlikely in this group of patients therefore the chance of the positive effect is even smaller.
No information on possible comorbidities of individuals in the analyzed studies is provided. This can greatly influence obtained data.
The title is unclear and should be changed.
Author Response
We would like to thank the reviewer for her/his thoughtful comments and efforts towards improving our manuscript.
The point-to-point response was included below:
The manuscript entitled „Body mass index reduction and cardiometabolic risk factors in obstructive sleep apnea reached by modified lifestyle: meta-analysis” provides information on available data regarding the influence of weight loss on AHI and chosen cardiometabolic parameters.
OSA is not a cardio-metabolic disorder – it is a sleep-breathing disorder affecting the cardiometabolic parameters of the individuals.
Thank you very much for this comment. Of course we agree on that and we focused on selected cardiometabolic parameters that could be affected by OSA.
The prevalence of the disorder is out-of-date. Two references given for the prevalence do not provide only cite given prevalence. If original publication were considered by the authors, they would have seen that it has been published in 1993. Present data suggest that it affect a vastly greater population (see Heinzers’ and Sentras’ papers).
Thank you very much for this comment. The prevalence was corrected according to data published by Heinzers’
Please expanded on the weight loss effect on AHI in the introduction, or give additional references, as the given one (no. 6) is from 1985 and this study included 23 patients.
Thank you very much for this comment. It was changed and expanded as well.
While discussing, the cardiometabolic risk factors in OSA, the frequency of its comorbidities from this group should be included.
Thank you very much for this comment. The frequency of its comorbidities was added.
Authors should undertake the topic that lifestyle change is very unlikely in this group of patients therefore the chance of the positive effect is even smaller. No information on possible comorbidities of individuals in the analyzed studies is provided. This can greatly influence obtained data.
Thank you very much for this comment. The information on „possible” lifestyle changes in this group of patients was added. Unfortunately authors followed inclusion and exclusion criteria from the study and did not report comorbidities in the papers. It is also not easy to contact them to get additional information. It was also added in the limitations.
The title is unclear and should be changed.
Thank you very much for this comment. The title was modified.
Reviewer 3 Report
This study is a meta-analysis that evaluates how BMI may influence AHI and Cardio-metabolic Risk in Sleep apnea patients.
This topic has a scientific relevance and, to my knowledge, few articles reported this association in this methodological way. Despite the limitations presented and expressed also by the same authors, I think that this article represents a good starting point that resumes the current knowledge about this theme. The paper, in fact, is well written and statistical analysis, table and figures, are clear.
I have only few minor corrections to suggest:
- In "Introduction" (about 9 line) I suggest changing the phrase “It classifies OSA into patients [4]” with “This latter defined four grades of OSA: mild (5.0-14.9), moderate (15.0-29.9), and severe (≥30.0 events per hour)”
- In Table 1 it is not defined the acronyms of VLED.
Author Response
We would like to thank the reviewer for her/his thoughtful comments and efforts towards improving our manuscript.
The point-to-point response was included below:
This study is a meta-analysis that evaluates how BMI may influence AHI and Cardio-metabolic Risk in Sleep apnea patients.
This topic has a scientific relevance and, to my knowledge, few articles reported this association in this methodological way. Despite the limitations presented and expressed also by the same authors, I think that this article represents a good starting point that resumes the current knowledge about this theme. The paper, in fact, is well written and statistical analysis, table and figures, are clear.
Thank you for very much for this comment.
I have only few minor corrections to suggest:
In "Introduction" (about 9 line) I suggest changing the phrase “It classifies OSA into patients [4]” with “This latter defined four grades of OSA: mild (5.0-14.9), moderate (15.0-29.9), and severe (≥30.0 events per hour)”
Thank you very much for this comment. It was changed.
In Table 1 it is not defined the acronyms of VLED.
Thank you very much for this comment. It was explained.
Round 2
Reviewer 2 Report
The authors have addressed the comments very well.
I think that dissuasion could also mention that at the moment more often more personalised attention to patient is present and more focused on different phenotypes of OSA are recognised such as REM-dependent phenotype (doi: 10.1038/s41598-019-56478-9) or positional phenotype (doi: 10.5664/jcsm.7166), which in future studies also should be considered while analysing effect of diet on cardio metabolic factors.
Author Response
We would like to thank the reviewer for her/his thoughtful comments and efforts towards improving our manuscript. The point-to-point response was included below:
I think that dissuasion could also mention that at the moment more often more personalised attention to patient is present and more focused on different phenotypes of OSA are recognised such as REM-dependent phenotype (doi: 10.1038/s41598-019-56478-9) or positional phenotype (doi: 10.5664/jcsm.7166), which in future studies also should be considered while analysing effect of diet on cardio metabolic factors.
Thank you for this comment. The suggestion done by reviewer was included and both papers were added to the reference list.